# Pentraxin 3 (PTX3): A Molecular Marker of Endothelial Dysfunction in Chronic Migraine

**DOI:** 10.3390/jcm9030849

**Published:** 2020-03-20

**Authors:** Clara Domínguez-Vivero, Yago Leira, Ana López-Ferreiro, Marta Saavedra, Xiana Rodríguez-Osorio, Tomás Sobrino, Francisco Campos, José Castillo, Rogelio Leira

**Affiliations:** 1Headache Unit, Department of Neurology, University Clinical Hospital, Universidade de Santiago de Compostela, 15706 Santiago de Compostela, Spain; clara.dominguez-vivero@gbhi.org (C.D.-V.); ana_loop@hotmail.com (A.L.-F.); marta.saavedra.pineiro@gmail.com (M.S.); xiana.ro@gmail.com (X.R.-O.); 2UCL Eastman Dental Institute and NIHR UCLH Biomedical Research Centre, University College London, London WC1X 8LD, UK; y.leira@ucl.ac.uk; 3Medical-Surgical Dentistry (OMEQUI) Research Group, Health Research Institute of Santiago de Compostela, 15782 Santiago de Compostela, Spain; 4Clinical Neurosciences Research Laboratory, Health Research Institute of Santiago de Compostela, 15706 Santiago de Compostela, Spain; tomas.sobrino.moreiras@sergas.es (T.S.); francisco.campos.perez@sergas.es (F.C.); jose.castillo.sanchez@sergas.es (J.C.)

**Keywords:** PTX3, sTWEAK, endothelial dysfunction, chronic migraine, FMD

## Abstract

Even though endothelial dysfunction is known to play a role in migraine pathophysiology, studies regarding levels of endothelial biomarkers in migraine have controversial results. Our aim was to evaluate the role of pentraxin 3 (PTX3) and soluble tumour necrosis factor-like weak inducer of apoptosis (sTWEAK) as potential biomarkers of endothelial dysfunction in chronic migraine (CM). We performed a case-control study including 102 CM patients and 28 control subjects and measured serum levels of markers of endothelial dysfunction (PTX3 and sTWEAK) and inflammation [high-sensitivity C-reactive protein (hs-CRP)] as well as brachial artery flow-mediated dilation (FMD) during interictal periods. Interictal serum levels of PTX3 and sTWEAK were higher in CM patients than in controls (1350.6 ± 54.8 versus 476.1 ± 49.4 pg/mL, *p* < 0.001 and 255.7 ± 21.1 versus 26.4 ± 2.6 pg/mL, *p* < 0.0001; respectively). FMD was diminished in CM patients compared to controls (9.6 ± 0.6 versus 15.2 ± 0.9%, *p* < 0.001). Both PTX3 and sTWEAK were negatively correlated with FMD (r = −0.508, *p* < 0.001 and r = −0.188, *p* = 0.033; respectively). After adjustment of confounders, PTX3 remained significantly correlated to FMD (r = −0.250, *p* = 0.013). Diagnosis of CM was 68.4 times more likely in an individual with levels of PTX3 ≥ 832.5 pg/mL, suggesting that PTX3 could be a novel biomarker of endothelial dysfunction in CM.

## 1. Introduction

Migraine is a neurological disorder that involves vascular and neural mechanisms in its pathophysiology. It is generally recognized that the development of a migraine headache depends on the activation of sensory afferent fibers of the trigeminal nerve, although the mechanisms leading to this activation are still unclear [1]. Trigeminal activation results in the release of inflammatory vasoactive peptides that promote vasodilation of the meningeal vessels and modulate endothelial function. The inflammatory response represents the key piece of evidence behind the role of endothelial dysfunction in the pathophysiology of migraine [2].

A bulk of evidence points to an association between migraine and endothelial dysfunction. Whether this association is causal or on the contrary—that endothelial dysfunction appears as a consequence of the disease—remains unclear [3]. The possible association between stroke or some non-atherosclerotic vascular conditions and migraine supports a role of endothelium in the etiopathogenesis of migraine. A systematic review [4] has recently analysed the relationship between several vascular biomarkers and migraine, concluding that there are not clear changes in their levels in patients. Promising biomarkers demand further investigation in the migraine population, for instance, brachial artery flow-mediated dilation (FMD). This technique has emerged as the most widely used non-invasive tool to assess endothelial function [5]. Previous FMD studies in migraineurs during interictal and ictal periods have shown contradictory results [6,7,8]. A study performed specifically in patients with chronic migraine (CM) showed diminished FMD [9].

Information regarding the association between CM and molecular biomarkers of vascular inflammation is limited. Increased plasma levels of fibrinogen and CRP were observed in patients with CM [9]. Pentraxin 3 (PTX3) is a member of the long pentraxin family that acts as an acute phase inflammatory glycoprotein [10]. Two studies have demonstrated higher plasma levels of PTX3 in migraine patients during attacks when compared to interictal periods [11] or healthy controls [12]. Plasma levels of soluble tumour necrosis factor-like weak inducer of apoptosis (sTWEAK) have been analysed as potential biomarkers of cardiovascular disease and endothelial dysfunction in vascular [13,14,15] and non-vascular diseases [16]. Our group demonstrated higher levels of PTX3 and sTWEAK in patients with severe periodontitis and CM [17]. Furthermore, we demonstrated that PTX3 and sTWEAK are elevated in CM patients and that high plasma levels of PTX3 can predict a good response to OnabotulinumtoxinA (OnabotA), which is widely used for the treatment of CM [18]. 

The aim of this study, therefore, was to investigate the relationship between levels of PTX3 and sTWEAK and FMD in order to elucidate their potential role as markers of endothelial dysfunction in CM patients during interictal periods.

## 2. Experimental Section

Subjects were recruited prospectively from the outpatient Headache Clinic of Department of Neurology at the Clinical University Hospital of Santiago de Compostela. One hundred and two individuals diagnosed of CM according to International Classification of Headache Disorders, 3rd edition criteria [19], were selected. Preventive treatment use was allowed. Twenty-eight healthy subjects without history of migraine or other type of headache were enrolled as a control group. Control subjects were recruited from the hospital and university staff, students, and general population. All subjects were older than 18 years old. Clinical variables were recorded, and selected molecular makers were determined in peripheral blood.

Exclusion criteria included the following: 1) high blood pressure (known high blood pressure or >2 measurements greater than 140/90 mm Hg); 2) coronary disease (coronary lesions > 50% determined by angiography, myocardial infarction, angina pectoris, or coronary recanalization); 3) diabetes mellitus (known diabetes mellitus or >2 fasting serum glucose determinations > 126 mg/dL); 4) hypercholesterolemia (pharmacologically treated or fasting serum cholesterol > 200 mg/dL); 5) infectious diseases; 6) chronic inflammatory conditions; 7) severe systemic diseases; 8) oligomenorrhea, polymenorrhea, or polycystic ovarian syndrome; 9) pregnancy or lactation; 10) obesity (body mass index > 30 kg/m2); 11) smoking habit (within the previous 12 months); 12) recent or chronic consumption of vasoactive drugs (>4 times the medium half-life of the active substance), including angiotensin-converting enzyme inhibitors and angiotensin receptor blockers. No patient was receiving preventive treatment for any of these conditions.

Written informed consent was obtained from each subject included in the study. The Research Ethics Committee of the Clinical University Hospital of Santiago de Compostela (Spain) approved the study (ID NO.: 2016/085). All procedures performed in the study were in accordance with the 1964 Helsinki declaration and its later amendments or comparable ethical standards.

All subjects filled a complete medical record including demographic data (age, gender) and personal and family history. Physical examination and clinical results were recorded. For migraineurs, type of migraine (with or without aura), time of evolution of the migraine (measured in years), intensity of headaches (measured by the visual analogic scale [VAS]), duration of attacks (quantified in hours), frequency of headaches (number of days with pain per month) as well as allodynia (>2 points on the 12 item allodynia symptom score, ASC-12) were registered. Clinical parameters were considered as an average of the patient’s episodes in the previous three months. We recorded possible comorbid conditions associated with migraine as well as symptomatic and preventive treatments for migraine.

After a screening visit, eligible subjects were invited to perform an ultrasonographic examination and blood sample extraction. Patients were headache-free from the previous 24 h to the visit. If a migraine occurred within the following 24 h, measurements were repeated in another headache-free period. Subjects had not previously consumed anti-inflammatory or analgesic medication in the previous 24 h to the blood sample extraction. Treatment with prophylactic drugs, when it existed, was not interrupted to perform the study. Measurements for control subjects and patients were performed between 10:00 and 11:00 am in a quiet, temperature-controlled room (22–24 °C) by a single observer. Subjects were at rest in the supine position for the previous 10 min. 

A blood sample was extracted from the non-dominant forearm. Blood samples were collected in chemistry test tubes, centrifuged at 3000 ×*g* for 15 min, and immediately frozen and stored at −80 °C. Serum levels of PTX3 and sTWEAK (Assay Biotech, Sunnyvale, CA, USA) were measured using commercial ELISA kits following manufacturer instructions. High sensitivity C-reactive protein (hs-CRP) was measured with an immunodiagnostic IMMULITE 1000 System (Siemens Healthcare Global, Los Angeles). The intra-assay and inter-assay coefficients of variation for all molecular markers were <8%. Determinations were performed in a laboratory blinded to clinical data.

After blood collection, the ultrasonographic study was completed in the dominant forearm. FMD of the brachial artery was assessed in all patients by the same researcher (A.L.-F.). The researcher was blinded to biochemical and molecular determinations and underwent previous technical training and validation of data (compared with medical staff skilled in neurosonology). We used a high-resolution B-mode ultrasound device (Aplio 50 Toshiba SSA-700) with a 7.5-MHz linear array transducer. FMD evaluations were performed according to the International Brachial Artery Reactivity Task Force and the Working Group of the European Society of Hypertension guidelines [20]. The dominant brachial artery was imaged 3–5 cm proximal to the antecubital fossa in a longitudinal plane, perpendicular to the ultrasound beam. Baseline measurements were first performed (d1, as the mean of 5 artery diameter determinations during systole) and location marked, followed by a rapid inflation of a cuff placed around the proximal forearm to 300 mm Hg for 4 min. Then a new determination was performed (d2, as the mean of new 5 determinations of the artery diameter during systole) 45–60 s after cuff release causing a reactive hyperemia. Brachial artery diameters were obtained from the near-to-far blood wall intima-media interfaces. FMD was expressed as the percentage of increase in the diameter from baseline (d2-d1/d1 ×100).

A formal sample size calculation was not done. However, considering a minimum expected effect size of 874.5 ± 108.0 pg/mL and including 102 CM cases and 28 controls, a post hoc power analysis calculation using the Macro !NSize for PASW Statistics (http://www.metodo.uab.cat/macros.htm.) was carried out, showing that our study had a power of 96% with an alpha risk of 5% to demonstrate significant differences between chronic migraineurs and healthy controls regarding the primary outcome of the study (i.e., PTX3 serum levels). Mean values ± standard error (SE) and median (P_25_, P_75_) were calculated for normally and non-normally distributed continuous variables, respectively. Statistical tests used to compare continuous data were the independent t-test or Mann-Whitney U test. Categorical variables were reported as percentages and compared by chi-square test. Analysis of covariance (ANCOVA) was used to create adjusted models using age, gender and BMI as covariates to compare mean values of FMD and biomarkers between cases and controls. Non-parametric correlation analysis between FMD, biomarkers and clinical variables was performed using Spearman’s rank correlation coefficient. In addition, partial correlations adjusted for same common confounders were also performed for FMD and significant biomarkers. The area under the Receiver Operating Characteristic (ROC) curve was performed to calculate a cut-off point for PTX3 in order to discriminate participants with and without CM. Logistic regression analysis was conducted to test the association between PTX3 (categorized according to calculated cut-off point) and diagnosis of CM adjusted for potential confounders (age, gender and BMI). All tests were carried out at a significance level of α = 0.05 using IBM SPSS Statistics (version 24.0, IBM Corp., Armonk, NY, USA).

## 3. Results

Baseline characteristics of the study population are shown in Table 1. No significant differences were found between CM patients and controls in terms of age, sex or BMI. Regarding migraine-related variables, almost half of CM patients presented aura. Allodynia was present in 32.4% of subjects. The average number of headache days in CM patients was 17.5 days per month with a mean duration of 21 h for each migraine attack. Intensity was recalled as high for most attacks. The number of years since first migraine symptoms was on average 16. 

FMD was significantly diminished in patients with CM compared to controls (9.6 ± 0.6 versus 15.2 ± 0.9, *p* < 0.001) (Figure 1A). Higher circulating levels of PTX3 and sTWEAK were found in CM compared to those without CM (1350.6 ± 54.8 versus 476.1 ± 49.4 pg/mL, *p* < 0.001 and 255.7 ± 21.1 versus 26.4 ± 2.6 pg/mL, *p* < 0.001; respectively) (Figure 1B and Figure 1C). These differences were confirmed in the multivariate model after adjusting for age, gender and BMI. No significant differences were observed for hs-CRP between groups (0.1 ± 0.03 mg/dL versus 0.3 ± 0.05 mg/dL, *p =* 0.178) (Figure 1D).

Both PTX3 and sTWEAK were negatively correlated with FMD (r = −0.508, *p* < 0.001 and r = −0.188, *p* = 0.033; respectively) (Figure 2A,B). No significant correlations were found between FMD, PTX3, sTWEAK and clinical or biochemical parameters (Table 2). After adjustment of confounders, PTX3 remained significantly correlated to FMD (r = −0.250, *p* = 0.013) while adjusted correlation between sTWEAK and FMD was attenuated, losing statistical significance (r = 0.005, *p* = 0.962).

The ROC (receiver operating characteristic) analysis for PTX3 had an area under the curve of 0.928 (95% CI: 0.884–0.971, *p* < 0.001), which suggests that this approach could discriminate between those patients with and without CM. The cut point of 832.5 pg/mL of PTX3 produced the optimal sensitivity and specificity of 83% and 85%, respectively and 86.9% of the participants were correctly assigned overall with an OR_adjusted_ of 68.4 (which means that CM is 68.4 times more likely in an individual with serum levels of PTX3 ≥ 832.5 pg/mL).

## 4. Discussion

In the present study, CM patients showed altered FMD (a direct measure of endothelial function) compared to healthy controls. Levels of PTX3 and sTWEAK were higher in CM patients and PTX3 levels were negatively correlated with FMD, even after adjustment for potential confounders. PTX3 serum levels ≥ 832.5 pg/mL independently predicted diagnosis of CM with a sensitivity of 83% and a specificity of 85%. Our results suggest that PTX3 could be a potential biomarker of endothelial dysfunction in CM.

The role of gender and sexual hormones in migraine and migraine chronification has been extensively studied [21]. Several studies have also evaluated the relationship between BMI and migraine. A higher risk for migraine has been reported in obese subjects and specifically for chronic migraine either in obese or pre-obese subjects [22]. This association is likely mediated by gender. In our study there were no differences among groups regarding sex or BMI, and differences observed in PTX3 levels survived multivariate analysis considering these factors.

To date, several studies have focused on endothelial dysfunction in migraine, however, their findings are controversial. Related molecular markers such as endothelin-1 (ET-1) or nitric oxide (NO) have been determined in migraine [23,24]. A systematic review has evaluated more than 600 reports measuring levels of biomarkers of inflammation, prothrombotic state, endothelial activation and endothelial repair [4] in migraine and concluded that results are conflicting and there are not definite vascular biomarkers in migraine patients. In the following years, novel potential markers of endothelial dysfunction were brought up and evaluated, such as endothelial microparticles (EMPs) [25], asymmetric dimethylarginine (ADMA) [26,27], and endothelial progenitor cells (EPCs) [28,29]. These molecules and cells play a role in endothelial regeneration and repair of injured vessels. However, to date, none of these biomarkers have been proven to be associated with vascular endothelial dysfunction specifically in migraine patients.

FMD is a reproducible and non-invasive ultrasound assessment of endothelial function [5]. Many studies assessing endothelial function in migraineurs have used FMD, however, their findings have been inconsistent [30,31,32]. In a previous study performed by our group, we did not find significant differences in FMD between control subjects and patients with episodic migraine (EM) during interictal and ictal periods. In the present study, however, we studied FMD in CM and found that it was significantly diminished when compared to controls, in agreement with a previous report [9]. FMD shows a high variability, and single determinations may reflect a transitory response of the endothelium more than a long-term process of endothelial dysfunction. Recurrent migraine attacks could influence persistent damage to the endothelium, explaining the different findings between CM and EM patients.

PTXs, a superfamily of soluble, multifactorial, pattern recognition proteins [33,34], are classical mediators of inflammation and markers of acute-phase reaction. PTXs are an essential component of the humoral response in innate immunity. In addition to CRP, the PTX superfamily includes the long PTX3, which is emerging as a key player in immunity and inflammation [35]. Unlike CRP, which is primarily synthesized in the liver, PTX3 is released by vascular endothelial cells and macrophages after stimulation with inflammatory cytokines (IL-1β, TNF-α), TLR agonists and microbial components. Therefore, PTX3 levels are believed to be a true independent indicator of local inflammation and are thought to reflect endothelial dysfunction more accurately than CPR [12]. Its concentration increases rapidly in various infections and plays an important role in the early phase of inflammation. PTX-3 can be used as a prognostic biomarker in sepsis and septic shock in adults [36] and children [37]. PTX3 plasma levels are also increased in some non-infectious conditions that involve inflammation and endothelial damage, such as vascular atherosclerosis, inflammation or vascular damage [38,39,40,41], especially after myocardial infarction [42], reflecting the extent of tissue damage. PTX3 has been associated with the incidence of coronary artery disease (CAD) and all-cause mortality in CAD patients [43]. Moreover, in patients with coronary artery disease, plasma levels of PTX3 have been correlated with endothelial function assessed by FMD showing a stronger association than the one existing between CRP and endothelial function [10], and it has been suggested that it might play a protective role in atherosclerosis [43].

Regarding migraine, only two studies have demonstrated higher plasma levels of PTX3 measured during attacks and compared to interictal periods [11] and control subjects [12]. To date, no information is available regarding peripheral blood biomarkers of vascular inflammation specifically in CM. Our group has reported higher levels of PTX3 and sTWEAK in patients with CM and concurrent severe periodontitis [17] as well as higher levels of PTX3 and sTWEAK in CM patients. On top of that, we found that PTX3 could be a predictor of a good response to OnabotA, which is widely used for the treatment of CM [18]. The present study follows this line of research and shows that plasma PTX3 is significantly correlated with endothelial function assessed by FMD in patients with CM, pointing to this as the most probable underlying mechanism. Furthermore, PTX levels ≥ 832.5 pg/mL predicted diagnosis of CM, thus indicating the potential role of PTX3 as a potential biomarker of endothelial dysfunction in CM.

TWEAK is a member of the TNF superfamily of cytokines that is synthesized as a type-II transmembrane protein from which a soluble form with biological activity can be released (i.e., sTWEAK) [44]. TWEAK mRNA is expressed in a variety of tissues and cells, including brain, heart, and lung, as well as human endothelial and smooth muscle cells [45]. TWEAK binds to Fn14, a highly inducible cell-surface receptor that is linked to several intracellular signalling pathways, including the nuclear factor-κ B (NF-κB) pathway [46]. TWEAK is a multifaceted cytokine whose effects are cell type and environment-dependent [47]. It may induce various cellular responses in vitro including cell proliferation [48], migration, differentiation [49], angiogenesis [48], and expression of pro-inflammatory molecules such as IL-8, MPC-1, ICAM-1, and E-selectin in human umbilical endothelial cells [50], and IL-6, IL-8, and ICAM-1 in astrocytes [51]. Plasma levels of sTWEAK have been evaluated as potential biomarkers of cardiovascular disease and endothelial dysfunction within several diseases. Low plasma levels of sTWEAK were associated with chronic vascular damage in carotid stenosis [13], coronary artery disease [52], peripheral artery disease [15], heart failure [14], and chronic kidney disease [45], and in patients with vascular risk factors such as diabetes [16], or hypertension [53]. In contrast, high plasma levels of sTWEAK have been described in patients after myocardial infarction [54] and stroke [55]. FMD has been associated with sTWEAK concentration in patients with chronic kidney disease [45]. The association of sTWEAK with FMD suggests that this protein could also be a biomarker of endothelial dysfunction.

In our study, we hypothesized that patients with migraine show increased levels of PTX3 and sTWEAK as a result of altered endothelial function, even in the absence of vascular risk factors. A recent review has found data supporting a relation between hypertension, transient ischemic attack (TIA), stroke, hypercholesterolemia and migraine, therefore we excluded subjects with the most prevalent vascular risk factors and other variables that may influence endothelium integrity and still found higher levels of PTX3 and sTWEAK in CM patients. Regarding systemic disease and its influence on CM, some chronic inflammatory conditions have been linked to migraine, such as endometriosis or gastrointestinal disorders. Systemic autoimmune diseases such as lupus, antiphospholipid syndrome and systemic sclerosis are more frequent in patients suffering from migraine, and endothelial dysfunction is the only alteration that is common among all these disorders. Considering that our exclusion criteria eliminated these confounders, it is tempting to postulate that high levels of PTX3 and sTWEAK are a result of pathophysiologic mechanisms related to migraine rather than a consequence of other conditions associated with endothelial dysfunction. These findings suggest a strong link between migraine, inflammation and endothelial dysfunction. Furthermore, no correlation was found between plasma levels of PTX3 and sTWEAK registered during interictal period and clinical parameters (intensity, duration of the headaches, time of evolution of migraine). These findings suggest that endothelial dysfunction could be a phenomenon related with the specific mechanisms underlying CM pathophysiology and not with the clinical variability of its manifestations.

This study has several limitations. First, the sample size was relatively small, particularly in the control group, and a larger population could have powered our results. FMD was studied only once in each subject when it is known to be variable between different measures, and the examiner who assessed FMD was not blinded for clinical data. The stage of the menstrual cycle was not taken into account and it has been proven to have an influence in endothelial function. Although our exclusion criteria support a relationship between endothelial dysfunction and migraine independently of cardiovascular cofounders, it also may limit the external validity of our findings.

## 5. Conclusions

In conclusion, patients with CM had increased circulating levels of PTX3 and sTWEAK as well as diminished FMD compared to subjects without migraine. PTX3 concentrations are negatively correlated with FMD. A value of PTX3 ≥ 832.5 pg/mL independently predicted the presence of CM. Hence, PTX3 could be considered as a potential biomarker of vascular endothelial dysfunction in CM. Further longitudinal studies are needed to confirm our preliminary results. Intervention studies are warranted to investigate whether a reduction in this biomarker could be translated into a beneficial clinical effect in chronic migraineurs.

## Figures and Tables

**Figure 1 jcm-09-00849-f001:**
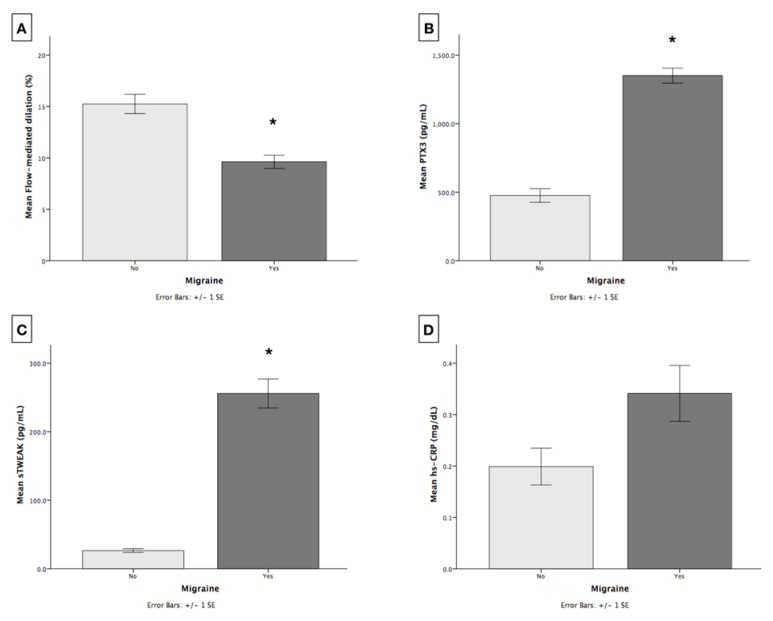
Comparisons between cases and controls for: (**A**) flow-mediated dilation (FMD) (%); (**B**) Serum levels of Pentraxin 3 (PTX3) (pg/mL); (**C**) Serum levels of soluble tumour necrosis factor-like weak inducer of apoptosis (sTWEAK) (pg/mL); (**D**) Serum levels of high sensitivity C-reactive protein hs-CRP (mg/dL). * *p* < 0.001.

**Figure 2 jcm-09-00849-f002:**
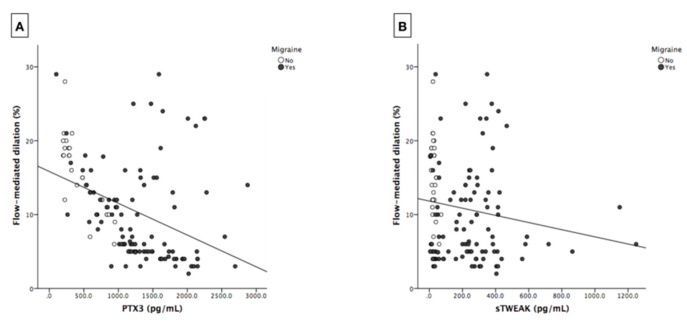
Scatter plots showing correlations between flow-mediated dilation (FMD) (%) and: (**A**) pentraxin 3 (PTX3) (pg/mL); (**B**) soluble tumour necrosis factor-like weak inducer of apoptosis (sTWEAK) (pg/mL).

**Table 1 jcm-09-00849-t001:** Baseline characteristics of the study population.

Variables	Controls (N = 28)	CM (N = 102)	*p*-Value
Age (years)	37.3 ± 1.6	39.5 ± 1.2	0.382
Gender, Females, *n* (%)	26 (92.9)	96 (94.1)	0.682
BMI (kg/m^2^)	24.3 (22.6, 25.9)	24.8 (22.3, 28.0)	0.474
Frequency of attacks (days/month)		17.5 ± 0.8	
Intensity of migraine attacks (VAS)		9.0 (8.0, 10.0)	
Duration of migraine attacks (hours)		21.3 ± 3.8	
Time of evolution of migraine (years)		15.9 ± 1.2	
Aura, *n* (%)		45 (44.1)	
Allodynia, *n* (%)		33 (32.4)	

CM: chronic migraine; BMI: body mass index; VAS: visual analogue scale.

**Table 2 jcm-09-00849-t002:** Correlations between flow-mediated dilation, clinical and laboratory parameters.

	Age	BMI (kg/m^2^)	Frequency	Intensity	Duration	Time of Evolution	hs-CRP (mg/dL)
**FMD (%)**	0.032	0.166	0.014	0.176	0.121	−0.048	0.156
***p*-value**	0.722	0.059	0.890	0.076	0.225	0.634	0.078
**PTX3 (pg/mL)**	−0.205	−0.096	0.29	−0.065	−0.125	−0.171	0.037
***p*-value**	0.190	0.276	0.771	0.516	0.211	0.085	0.676
**sTWEAK (pg/mL)**	−0.157	0.025	−0.118	−0.078	−0.118	−0.124	0.095
***p*-value**	0.075	0.779	0.236	0.437	0.236	0.216	0.283

BMI: body mass index; FMD: flow-mediated dilation; PTX3: pentraxin 3; sTWEAK: soluble fragment of tumor necrosis factor-like weak inducer of apoptosis; hs-CRP: high sensitivity C-reactive protein.

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
