# Peer review of "Pentraxin 3 (PTX3): A Molecular Marker of Endothelial Dysfunction in Chronic Migraine"

_jcm, 2020, doi:10.3390/jcm9030849_

Round 1
Reviewer 1 Report
This study presents data regarding novel migraine biomarkers in a large group of subjects with chronic migraine and controls. This study of and pentraxin 3 and sTWEAK adds to the existing evidence that endothelial dysfunction is either a cause or biomarker for migraine, particularly chronic migraine. Overall I think this concept is important and potentially significant for the field.
Some of the vascular biomarkers you mentioned (lipids, CRP, platelets, TNF-α, fibrinogen, IL-1, adiponectin) don’t seem especially specific or linked to migraine. I’d rather state that many putative vascular biomarkers you mentioned have not been clearly shown to be linked to migraine.
For the reader, it might be helpful to provide more background about the studies reviewing pentraxin-3 elevations in other disorders such as sepsis or CV disease. How long would you expect pentraxin-3 to be elevated during a migraine attack? Do serum levels accurately predict what’s going on in the brain? Feel free to elaborate.
It might have been helpful to recruit patient from other clinic populations to avoid Berkson’s bias when comparing healthy volunteers to patients at a University-based headache clinic.
Although I read the discussion about previous studies in episodic migraine, I was a little confused about why you decided to study CM specifically, and using controls but not episodic migraine with or without aura. As a potential biomarker – it could be useful to see if pentraxin 3 goes down with successful treatment, or if it could clarify diagnosis. Making a diagnosis of CM is not especially challenging, but it would fascinating to have biomarker to help measure disease severity and determine if treatments are working. As an example, the Rodríguez-Osorio 2012 Neurology paper didn’t suggest a link between endothelial dysfunction and migraine frequency/severity.
Other questions:
How did you determine the healthy controls did not have migraine?
Can you explain the reason for excluding “recent vasoactive drugs?” Did you consider triptans vasoactive drugs?
How did you determine subjects had allodynia? Did you use a validated questionnaire or test them for allodynia?
I’m not sure what you mean by “Treatment with prophylactic drugs, when existed, was not interrupted.” Wasn’t this study conducted in one visit – I don’t see where this is long-term study.
There is some evidence that angiotensin-converting enzyme inhibitors, angiotensin receptor blockers may alter endothelial function. As these medications are occasionally used for migraine prevention would it be better to exclude them from these studies?
Clarify: “almost half of CM patients presented aura and were also diagnosed of tension-type headache.” I believe you mean that almost ½ have aura. In terms of tension-type headache, given that CM patients by definition have at least 15 headache day/month and at least 8 migraine days, what does this mean?
Reviewer 2 Report
Dear authors,
I have read the mauscript and I send you my comments:
1) Methods: please indicate the power calculation
3) Results please clarify the role of BMI, gender and systemic diseases in the development of migraine;
4) Results: please add a figure documenting the plasma values of PTX3 in patients with migraine in the respect of the time od pain
5) Results: please add data of PTX3 modulation after drug use
6) Discussion must be rewritten considering the results
Round 2
Reviewer 2 Report
none